# Fear and Stigma of COVID-19 Reinfection Scale (FSoCOVID-19RS): New Scale Development and Validation

**DOI:** 10.3390/healthcare11101461

**Published:** 2023-05-18

**Authors:** Zainab Fatehi Albikawi, Mohammad Hamdi Abuadas, Mesheil M. Alalyani, Yousef Zahrani, Emad Aqel, Raid Safi

**Affiliations:** 1Community and Psychiatric/Mental Health Nursing Department, Nursing College, King Khalid University, Khamis Mushait 39746, Saudi Arabia; 2Medical Surgical Nursing Department, Nursing College, King Khalid University, Khamis Mushait 39746, Saudi Arabia; 3Public Health Department, College of Applied Medical Sciences, King Khalid University, Khamis Mushait 62529, Saudi Arabia; 4Basic Nursing Care Department, Nursing College, King Khalid University, Khamis Mushait 39746, Saudi Arabia

**Keywords:** fear, stigma, reliability, validity, reinfection, COVID-19, students, nursing

## Abstract

Background: The advent of COVID-19 and its impacts have prompted fear and stigma among people all across the world. Because of stigma, there was often a delay in diagnosis and treatment, which resulted in a poor prognosis. As a result, a reliable scale is required to measure the level of fear and stigma of COVID-19 reinfection. Aim: To develop and validate a scale for determining the level of fear and stigma of COVID-19 reinfection. Methods: A cross-sectional study including 200 nursing-college students who had previously tested positive for COVID-19 was conducted. The scale’s reliability was evaluated by external and internal consistency methods. Construct, convergent, and discriminant validity were evaluated using exploratory factor analysis (EFA) and confirmatory factor analysis (CFA). Results: The scale’s mean score was 24.85 ± 11.35, and no floor or ceiling effects were detected. The scale items’ reliability, measured by Cronbach’s alpha coefficient if an item was deleted, ranged from 0.76 to 0.95, with a total score value of 0.86. The range of convergent validity coefficients was between 0.37 and 0.64. Pearson’s correlation coefficients for test–retest validity ranged from 0.71 to 0.93, with a total score of 0.82. The coefficient of split-half correlation was 0.87, while the coefficient of reliability was 0.93. According to the factor analysis, two components had latent roots larger than 1. The rotated component matrix of the two factors revealed that all items had *R* values over 0.30, indicating that none of them should be excluded. In addition, CFA results revealed that χ^2^ = 3524, df = 1283, χ^2^/df ratio = 2.74, *p* < 0.001, GFI = 0.86, CFI = 0.92, AGFI = 0.88, and RMSEA = 0.05. The scale’s convergent and discriminant validity was confirmed. Conclusions: The 14-item, two-dimensional Fear and Stigma of COVID-19 Reinfection Scale (FSoCOVID-19 RS) was demonstrated to have reliable psychometric properties.

## 1. Introduction

In December of 2019, Wuhan, China, was the location where the first instances of a novel coronavirus disease known as COVID-19 were recorded. In March of 2020, it was determined that the disease had reached pandemic proportions. As of September 2020, the World Health Organization (WHO) had received reports of 209,201,939 confirmed cases of COVID-19 around the world, including 4,390,467 fatalities. The virus has a high rate of transmission from one person to another [1]. Individuals understandably started to worry about COVID-19 when the infection rate was so high and the fatality rate was also relatively high. According to some reports, people are afraid to come into contact with other individuals since they might have COVID-19 [2].

Over a thousand distinct clades and sub-clades of the virus have been discovered so far, as time goes on and efforts to catalog them persist. There may be a genetic shift in the virus as a result of natural selection favoring strains that are better able to evade the immunological responses that are triggered by primary infections [3]. In this hypothetical but conceivable scenario, reinfection unquestionably becomes a reality that cannot be avoided. Cases of reinfection further show that herd immunity against COVID-19 cannot develop with mere immunity originating from spontaneous infections alone [4]. Addressing the individual’s fears about reinfection, educating people about its likelihood and prevention techniques, and requesting ongoing monitoring regardless of the pandemic’s peaks and valleys are a few of the ways to ensure their compliance. Individuals should be made aware of the benefits of supporting testing and surveillance, maintaining strong social-distancing measures, mask compliance, and early immunization [5].

Existing pre-pandemic psychiatric problems or emotional discomfort may be exacerbated by the fear of infection, leading to acute anxiety [6]. According to the findings of studies, one of the symptoms of post-traumatic stress disorder is an excessive fear of COVID-19 [7,8]. Fear is one aspect that distinguishes infectious disease from other types of conditions due to its unique nature [9]. In most cases, fear serves a useful purpose and is associated with fundamental biological processes that help individuals prepare for survival. These processes enable them to react to potential dangers by taking protective action, such as escaping, isolating themselves, or distancing themselves [10]. Fear of COVID-19 also prompts individuals to remain on high alert in order to protect both themselves and the people they care about, which can result in feelings of social isolation, fear, and panic in certain individuals [11].

Stigma associated with COVID-19 refers to an unfavorable attitude towards those who are infected with the virus or have intimate contact with people who have the virus [12]. Recent research found that those who had recently been released from quarantine, people who were infected or suspected of having COVID-19, and people who had recently returned from overseas all experienced some form of stigmatization, in which they were socially excluded, humiliated, and stereotyped [13].

A study compared the levels of stigma experienced by COVID-19 survivors and healthy controls in China. The survivors reported higher levels of general stigma, which resulted in social isolation and feelings of shame [14]. In a further study conducted in Vietnam, it was discovered that 34% of healthcare professionals felt stigmatized, and as a result, they avoided relationships with neighbors and those living in the neighborhood. In addition, 10% of healthcare workers felt blameworthy by friends and family [15]. Furthermore, reports of health risks and psychological issues, such as depression and suicide, were recorded [16,17].

Due to the fact that people who are stigmatized are less likely to reveal their health status [13], they consequently avoid activities that include getting help from a professional [18] and taking the COVID-19 test [18]. The stigma associated with COVID-19 may potentially undermine attempts designed to contain and stop the pandemic from spreading. This is due to the fact that stigmatized individuals are less inclined to reveal their health status.

During times of pandemics, it is recommended that the mental health of students in colleges be monitored [19,20]. On the other hand, there is a paucity of research on the psychological effects that COVID-19 has on undergraduate students, particularly nursing students [20]. Individuals may experience fear as a result of pandemics and epidemics.

Nursing students in Saudi Arabia have a concern of contracting the COVID-19 virus at a rate of 79.3% [21]. In addition, many Turkish nursing students reported feeling fearful about contracting COVID-19 [22]. Fear of COVID-19 has been found to be moderate to high among nursing students in various studies [22,23,24,25]. According to research by Beisland, et al. [26], the COVID-19 fear levels of nursing students are significantly greater than those of the general population. According to national and international studies, the fear of COVID-19 is pervasive among nursing students. It was discovered that fear of COVID-19 increases students’ propensity to opt out of study [23]. In order to minimize adjustment issues, depression, irritability, anxiety, and anger, it is crucial to detect this fear as early as possible [27].

Therefore, it is essential to assess the levels of fear experienced by nursing students in order for them to develop effective coping mechanisms for dealing with these feelings [20,28,29]. It is not easy to deal with the stigma that was created as a result of the COVID-19 outbreak. Stigma brings up issues that are extremely complicated and affect people’s mental health. Because it might persist for a long period without being addressed by effective and efficient social initiatives, the stigma of COVID-19 is a problem that will have a significant impact on social life [30]. As a result, it is necessary to address this concern among nursing students.

When compared to the pre-pandemic baseline, reports of psychosocial distress among adolescents rose during the COVID-19 pandemic [31]. Nevertheless, the potential of the individual’s psychosocial components has not been completely examined. To achieve the broad goal of creating a society free from the psychosocial burden of COVID-19, the study of the fear and stigma of COVID-19 reinfection is essential for maintaining the community’s mental stability in the face of a pandemic. It is possible that the lack of a suitable psychometric scale is the reason why the current treatment for COVID-19 pays such little attention to the fear and stigma of COVID-19 reinfection. This is the only scale that we are aware of that measures fear and stigma among individuals who have a history of COVID-19 infection. The purpose of this research was to develop and validate a scale for determining the level of fear and stigma of COVID-19 reinfection.

## 2. Methodology

### 2.1. Design and Setting of the Study

The study was cross-sectional. The fact that this method of research has the potential to yield findings on a large number of participants in a relatively short length of time was a major aspect that played a role in the decision to carry out a cross-sectional study as the method of research in this study. The research was conducted among students at a nursing college in Saudi Arabia.

### 2.2. Population and Sample

Nursing-college students who were previously COVID-19-positive were included in the study sample. Until recently, the majority of sample size recommendations were based on simple concepts derived from the professional experience of experts. Some of the most frequently mentioned suggestions include absolute numbers. Gorsuch [32] and Kline [33] recommended a minimum sample size of 100 individuals. Comrey and Lee [34] established the following sample size adequacy scale: 50 (extremely poor), 100 (poor), 200 (medium), 300 (good), 500 (very good), and more than 1000 (excellent).

Moreover, the researchers in the current study used Epi-Info (Epidemiological Information Package) version 7.2.5.0 to determine the sample size based on the following information: There were 600 students in the college of nursing. In an earlier study, 79.3% of nursing students in Saudi Arabia stated they had a fear of COVID-19 infection [21]. So, with a 95% confidence interval, 178 cases were thought to be the required number for the sample. The a-priori Sample Size Calculator [35] also confirmed the sample size with a significance level of 0.05, an effect size of 0.25, statistical power of 0.85, and two latent variables. As a result, the minimum sample size for this study was determined to be 156 participants. To account for the low response rate, the researchers recruited additional participants. Additional participants equal to 20–30% of the predicted total are usually necessary [36]. That is why the current study’s researchers assumed that 200 participants would be sufficient for the sample size.

Participants needed to meet the following criteria in order to be considered for the study: (1) undergraduate nursing students who could read, understand, and communicate; (2) the students needed to have sufficient time; (3) the students needed to give their informed consent in order to participate willingly in the study; and (4) the students had to have previously been positive COVID-19, as shown in the Tawakkalna application, which is intended to deliver a wide variety of integrated services to the Saudi Arabian population in order to raise the country’s standard of living. There were pillars that made up the Kingdom of Saudi Arabia’s e-health response to COVID-19. The five mobile phone applications that made up those  pillars were Mawid, Tabaud, Sehha, Tetamman, and Tawakkalna. Only one of these applications—namely, the Tawakkalna App—was required to be used by the population of the Kingdom of Saudi Arabia. This mandate applied only to the Tawakkalna App [37]. Finally, students with pre-existing mental health disorders were not permitted to take part in the research study.

### 2.3. Development of the Scale

There was a multi-step process that had to be completed before the Fear and Stigma of COVID-19 Reinfection Scale (FSoCOVID-19RS) could be developed, including item generation and psychometric properties [38]. First, a comprehensive review of the relevant literature was performed with the intention of assessing each general component of the fear and stigma scales. There were thirty different items found that could be used to assess the fear and stigma of COVID-19 reinfection. These methods of assessing fear and stigma are applicable to a wide range of populations and conditions. The pool contained all of the pertinent and potentially useful items that had been combined.

After removing the items that contained material or expressions that were the same as those found in other items, the evaluation proceeded with the remaining 25 items. Secondly, an expert panel consisting of a psychotherapist, a health counselor, a psychiatrist, a general physician, and a psychiatric/mental health nurse examined all 25 items. Based on the recommendations made by the expert panel, 10 items were removed from the scale. Third, the 15 items that were kept were given to a separate group of experts so that they could evaluate them. Following the review of the second expert panel, one more item was scratched off the list and removed from consideration. On a scale from zero to three points, the following responses were given for each item: not at all applicable to me (0), applicable to me to some extent (1), applicable to me to a substantial degree (2), and applicable to me frequently (3). The total item scores from the FSoCOVID-19RS were summed to reflect the individual’s fear and stigma of COVID-19 reinfection level, and the resulting scores varied proportionally from 0 to 42, with a score of 0 indicating the absence of fear and stigma of COVID-19 reinfection and a score of 42 indicating the highest level of fear and stigma associated with COVID-19 reinfection.

### 2.4. Pilot Test

The purpose of the pilot study was to collect data on the first psychometric properties of the scale and to provide a platform for conducting a simulation of the research work. The expectations were that both of these goals would be accomplished. The pilot study consisted of ten percent of the total sample size, or twenty participants; however, these individuals were not included in the main study population.

### 2.5. Statistical Analysis

The researcher used a method called double-entry data entry to ensure that there were no mistakes made throughout the transcription process. The Statistical Package for the Social Sciences, version 25.0 (SPSS), was chosen [39]. After observing the floor and ceiling effects, it was decided that items with usual responses should not be included in the study if those responses were detected. This decision was made after the floor and ceiling effects were observed. Typical replies are those in which the majority of the responses are clustered at either the lowest possible scale score or the highest possible scale score. The responsiveness of the scale is negatively impacted as a result of this factor. To analyze construct validity, the convergent validity of the scale was employed. This was done to determine whether or not there was a correlation between items and the overall score [40]. The reliability of the scale was determined statistically with the Cronbach’s alpha method. Estimates of the coefficients were established, and the results obtained were considered satisfactory if they were greater than 0.70 [41]. In addition to this, the split-half reliability coefficient was applied in order to analyze the scale’s internal consistency. Each of the strategies that was described ought to have appropriate correlation coefficients that were greater than 0.30 [42]. Test–retest validity was applied in order to determine the extent to which the scale exhibited external consistency. In this particular research study, the participants were asked to do the survey not once but twice, with a two-week break in between each try; the researchers then compared the responses they gave on the first and second attempts to look for any associations [43].

It was determined through the utilization of the Kaiser–Meyer–Olkin test of sampling adequacy that the sample size was adequate. When evaluating the size of the sample, it is important to look for results that are greater than 0.50. It is considered statistically significant if the *p* value is less than 0.05, and the Bartlett test of sphericity was used to determine whether or not there was a statistically significant correlation between the items [44]. Utilizing exploratory factor analysis (EFA) allowed for the elimination of any poor items that were present on the scale. The extraction of the component factor was used in order to determine whether or not there were any factors that had a latent root that was larger than the one that was supposed to be used. In order to determine whether or not each factor contained a sufficient number of items, an extraction method based on principal component analysis and a rotation method based on Varimax with Kaiser normalization were utilized. It was determined that an item was powerful if its saturation was greater than 0.30 [21]. The two-factor structure that was taken from EFA was evaluated using confirmatory factor analysis (CFA) with the maximum likelihood estimation method in order to determine whether or not it possessed construct validity. Convergent and discriminant validity were assessed by EFA and CFA, including the average variance extracted (AVE) and the composite reliability (CR) of the scale.

## 3. Results

The study involved 200 students who had been identified previously as having COVID-19 and whose status had been verified in the Tawakkalna application. Students’ ages ranged from 18 to 26 years, with a mean age of 21.73 (SD: 11). One hundred and fifteen of the students (57.5%) were female, 26 (13.68%) had been diagnosed with a chronic illness, and 21 (11.05%) had a history of mental illness in their families. As can be seen in Table 1, the vast majority of the participants were single (73.68%), and 32.63% of the participants were in their third year.

When forms with typical replies (responses that were positioned at the lowest or maximum scale score), as shown in Table 2, were eliminated, the total number of participants decreased to 190. The scale score for these answers was either the lowest or the highest. These responses can be found in either the lowest or maximum possible scale score. On the Fear and Stigma of COVID-19 Reinfection Scale, calculations were done to calculate the mean (24.85), median (28), and standard deviations of the scores (11.35), as well as the minimums (0) and maximums (42) of those values. High levels of fear and stigma of COVID-19 reinfection were displayed as the participant’s total score increased. It was determined that the floor and ceiling effects were not present in the dimensions; more specifically, there were no responses that were higher than 15% found at the top or bottom of the scale; this indicates that the responsiveness of the scale was improved.

In order to test and evaluate reliability, the Cronbach’s alpha coefficient was applied. Table 3 presents the coefficients that would be obtained if the item in scale were removed, in addition to the overall alpha value of the scale dimensions. In the process of determining how removing each item would affect the overall alpha coefficient of the scale dimensions, the researchers found that the variances ranged from 0.76 (having COVID-19 again threatens my family’s reputation) to 0.95 (when I watch the news and read stories about the possibility of COVID-19 reinfection, I become scared) and 0.86 for the overall score variances ranged from 0.76 (having COVID-19 again threatens my family’s reputation) to 0.95 (when I watch the news and read stories about the possibility of COVID-19 reinfection, I become scared) and 0.86 for the overall score. Cronbach’s alpha was adequate because every one of the items’ scores was greater than 0.70, indicating that all of the items’ alpha ratings were acceptable.

In terms of the split-half reliability coefficient of the scale, the correlation coefficient was 0.87 and the reliability coefficient was 0.93; both of these correlation coefficients are regarded as satisfactory in terms of the internal consistency of the scale.

For the purpose of determining whether or not the scale possessed construct validity, convergent validity was examined. The different components of the scale and the overall score on the scale were analyzed using Pearson’s correlation coefficients so that researchers could better understand the relationship between the two. The convergent validity coefficients are illustrated in Table 4, which can be found below. The values ranged between 0.37 (I believe it is vital to conceal from others the fact that I have been reinfected) and 0.64 (When I watch the news and read stories about the possibility of a COVID-19 reinfection, I become scared). All of the items that were evaluated had correlation coefficients that were higher than 0.30, which is the level at which results are regarded as satisfactory. Correlation values above 0.25 between items were considered quite high by Bernstein [45].

To determine the extent to which the scale possessed external consistency, test–retest validity was utilized. The data were analyzed using Pearson’s correlation coefficients, and the comparison of the participants’ responses from the first time with those from the second time was done. The findings are summarized in Table 4. The range of the value was from 0.71 (when I think of COVID-19 reinfection, I get a bad feeling) to 0.93 (I have recurrent nightmares in which I recall my previous COVID-19 infection), with an overall score for the whole being 0.82. Each of the items that was analyzed had correlation coefficients of more than 0.30, which is the minimum value that may be regarded as acceptable. Kirk and Miller (1986) state that the scale’s validity is assessed by looking at the degree to which individual item scores correlate with the overall score. Each item’s score on a construct is compared to the construct’s total score using the Pearson correlation.

The Kaiser–Meyer–Olkin measure of sampling adequacy was utilized to determine that the sample size was adequate, and it was found to be 0.92. Additionally, the result of 0.97 for Bartlett’s test of sphericity and a *p* value of less than 0.001 imply a highly statistically significant link between the items. The results of the extraction of the component factor using the component principal on commonality transactions for the variable are displayed in Table 5. It was discovered that two factors had latent roots that were greater than 1, and these factors are shown below. The percentage of each factor’s variance that could be interpreted in terms of the total variance was determined. The first factor had a possible root value of 9.720 and accounted for 72.22% of the total variation. On the other hand, the potential root of the second factor was equal to 1.14, and it was responsible for explaining 80.25% of the total variance. Table 5 displays the findings of a rotated component matrix applied to the two factors, which revealed that the first factor explained 72.22% of the variance and included questions 3, 4, 6, 7, 9, 12, and 14, while the second factor explained 80.25% of the variance and included questions 1, 2, 5, 8, 10, 11, and 13. Because the correlation coefficients of each question were greater than 0.30, it was not necessary to eliminate any of the items from consideration.

### 3.1. Confirmatory Factor Analysis (CFA)

CFA [46] was utilized in order to examine the scale’s two-factor structure. The findings showed that a two-factor structure was the most appropriate framework for the 14-item scale. The path diagram illustrating the CFA of standardized regression for the 14 items on the Fear and Stigma of COVID-19 Reinfection (FSoCOVID-19RS) can be found in Figure 1. The factor loadings (standardized regression) for the latent variables (circles) and each of the observed variables (rectangles) are shown in the route diagram. Additionally, the amount of variance (R2) that the latent variables account for in the observed variables is shown. Schermelleh-Engel, et al. [47] suggested thresholds for an acceptable fit, which were used as criteria to evaluate how well the model fit the data. The following are the criteria that were employed: (a) the critical ratio (CR) of the factor loadings should be greater than 1.96, (b) the index of relative chi-square (χ^2^/df) should be less than 5, (c) the comparative fit index (CFI) and the normed fit index (NFI) should both be more than 0.85, (d) the goodness of fit index (GFI) and the adjusted goodness of fit index (AGFI) should both be more than 0.85, and (e) the standardized root mean square residual (RMR) and the root mean square error of approximation (RMSEA) should both be less than 0.08. In this analysis, all of the factor loadings were found to be between 0.68 and 0.86, and the correlation coefficient was determined to be greater than 1.96, suggesting that the results were statistically significant.

The fit indices for the 14-item FSOCOVID-19RS were as follows: χ^2^ = 3524, df = 1283, χ^2^/df ratio = 2.74, *p* < 0.001, GFI = 0.86, CFI = 0.92, AGFI = 0.88, and RMSEA = 0.05. RMSEA stands for root mean squared error. The great majority of the findings indicated that the model had a good enough fit. Following that, modification indices were investigated to determine the parameters that indicated cross-loadings and misspecifications. Nonetheless, no changes were made because no significant improvement in fit indexes was identified. The model was accepted in its existing form, despite its complexity.

### 3.2. FSoCOVID-19RS Scale Convergent and Discriminant Validity

The degree to which items for a particular construct share a high proportion of the variance is what is meant by the term “convergent validity”, whereas the degree to which a construct is actually unique from other constructs is what is meant by the term “discriminant validity” [48]. Composite reliability (CR), average variance extracted (AVE) (which represents the average percent of variation explained among the items in the same construct), maximum shared variance (MSV), and average shared variance (ASV) are some of the measures that can be used to establish convergent and discriminant validity. Convergent validity was determined by comparing each construct’s AVE to its association with the other constructs; convergent validity was proven when AVE was greater than 0.50 [48]. MSV and ASV should be below AVE for all constructs to show discriminant validity [48]. AVE scores for all dimensions on the FSoCOVID-19RS scale were over 0.50 and greater than correlations with other items, demonstrating the scale’s convergent validity. MSV and ASV were found to be lower than the AVE for all scale constructs, demonstrating the discriminant validity of the FSoCOVID-19RS scale (see Table 6).

## 4. Discussion

Examining the psychometric features and developing a scale for determining the level of fear and stigma of reinfection with COVID-19 among nursing-college students who were previously infected with COVID-19 was the purpose of this study. The scale is innovative and will be useful for future research, be applicable in various cultures, and help in reducing the detrimental effects of stigma and fear of reinfection.

The scoring system was simplified so that it is straightforward, clear, and easy to use. Because of its four-point style, the Likert scale made it possible for respondents to provide a wide variety of responses. It addresses the fear that is associated with reinfection, which is represented by items 3, 4, 6, 7, 9, 12, and 14, and it also addresses the stigma that is associated with reinfection, which is represented by items 1, 2, 5, 8, 10, 11, and 13.

There were no floor or ceiling effects, and there was extremely good reliability (Cronbach alpha range of 0.76–0.95 and 0.86 for the total score) and strong convergent validity (*R* range of 0.37–0.64); therefore, the scale is regarded as a very good instrument for measuring fear and stigma of reinfection with COVID-19. There was also strong internal consistency (split half correlation *R* was 0.87 and reliability *R* was 0.93) and strong external consistency (test–retest *R* ranged from 0.71 to 0.93 and 0.82 for the total score). The fact that all of the items on the scale had *R* values more than 0.30, which indicates that there is no requirement to eliminate any of them, is a strength of the scale. In addition, the findings of the CFA showed that a two-factor structure was the most appropriate framework for the 14-item scale.

Because the pandemic has had such a pervasive influence on the mental health of the general population, there has been a growing demand for measures that can evaluate the various facets of fear associated with COVID-19 [49]. Studies have identified various domains of fear related to COVID-19 infection, such as fear of oneself or their family becoming infected; fear of economic losses and unemployment; fear of, or avoidance behaviors towards, gaining knowledge about the pandemic; and fear of deciding whether to visit parents or not or whether to seek information on death rates [50,51]. It is possible that people’s reactions to control guidelines that are necessary as preventative measures to help with the overall outcome of disease transmission in the community will also be influenced by this fear [52]. Fear of contracting the COVID-19 virus has caused a great number of people to give up participating in social activities [53]. Because of the pandemic’s direct or indirect consequences, younger people, such as college students or adolescents, were particularly susceptible to issues related to their mental health during the pandemic. Students who are fearful of contracting COVID-19, who are pessimistic about their chances of getting it, or who have family or friends who have been infected are more likely to develop anxiety or depression [54]. Because of the severity, mortality, and susceptibility of the diseases, college students may experience an increase in their level of fear, which may have an impact on their mental health as well as their ability to learn well [55,56].

A person who is stigmatized is more likely to be the target of harassment, bullying, discrimination, and the breaking of social links and relationships. In addition to this, there have been reports of health risks and psychological issues, such as depression and suicide [16,17]. A study was conducted on households that had at least one confirmed case of COVID-19. The researchers found that these families were reluctant to disclose their COVID-19 status to others and to meet others after they had been quarantined and isolated because they feared being stigmatized. They even refrained from addressing their concerns and misgivings regarding COVID-19 with members of their families [57]. Some people who survived COVID-19 said that they were even avoided by their neighbors and coworkers [58].

### 4.1. Limitations

The current study did have certain limitations. First, despite the fact that fear and stigma are personal experiences and that there are limitations when attempting to evaluate them objectively, the nature of the self-report cannot rule out the possibility that respondents’ answers were influenced by social desirability factors. This is because fear and stigma are personal experiences. Second, there were no reverse-coded items included in the scale. Finally, the present study’s findings are less generalizable due to the use of a convenience sample of students from a nursing college.

### 4.2. Implications

The COVID-19 pandemic has been significant in bringing attention to the various factors that influence psychosocial health. The current study focused how to measure the fear and stigma of reinfection with COVID-19 among individuals who have previously been infected with the virus. During times of pandemic, it is crucial to keep an eye on the psychosocial health of the population since it has become increasingly clear that fear and stigma might be among the most significant factors that contribute to a decline in mental and psychosocial health. The implications of the current study relate to the conduct of additional psychometric testing as well as the use of the tool in studies testing the efficiency of educational programs on the level of stigma and fear of reinfection.

## 5. Conclusions

The Fear and Stigma of COVID-19 Reinfection Scale (FSoCOVID-19RS) is a 14-item two-dimensional scale that was shown to have reliable psychometric features. It is the first scale of its kind to measure both fear and stigma associated with reinfection with COVID-19 among previously infected individuals. It has the ability to promote the advancement of operational research and the development of ways to eliminate the stigma associated with COVID-19 as well as the fear of reinfection. It just has to be translated into the appropriate language for that particular culture. Further research on the elements that contribute to stigma and fear in society is strongly encouraged.

## Figures and Tables

**Figure 1 healthcare-11-01461-f001:**
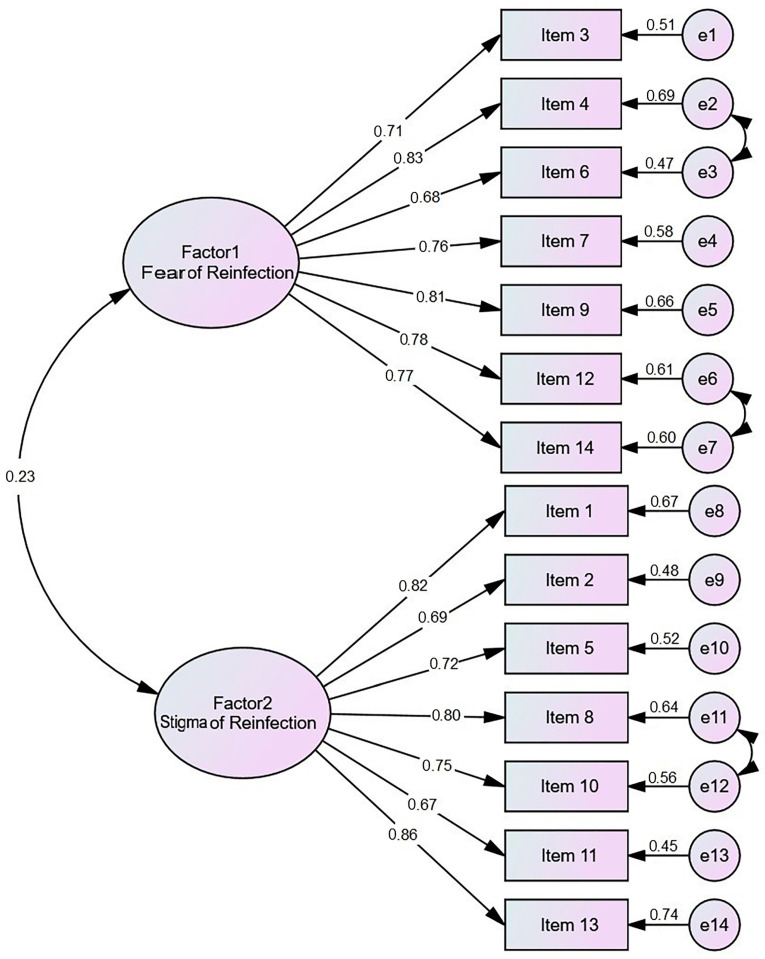
Path diagram illustrating the CFA and regression standardization for FSoCOVID-19RS (*N* = 190).

**Table 1 healthcare-11-01461-t001:** Sociodemographic details of the group under study (*N* = 200).

Variable	Number	Percentage
Age (years)		
18–22	150	75.00
≥23	50	25.00
Gender		
Male	75	37.50
Female	115	57.50
Marital status		
Married	50	26.32
Single	140	73.68
Educational Level		
1st year	40	21.05
2nd year	56	29.47
3rd year	62	32.63
4th year	22	11.58
Internship	10	5.27
Chronic disease diagnosis		
Yes	26	13.68
No	164	86.32
Mental illness in the family		
Yes	21	11.05
No	169	88.95

**Table 2 healthcare-11-01461-t002:** Mean, median, range, and ceiling and floor effects.

	Total Mean (SD)	Median	Minimum	Maximum	
Number of participants = 190	24.85(11.35)	28	0	42	
				No.	%
	Floor effect (responses receiving the lowest rating)			7	3.5%
	Ceiling effect (responses receiving the highest rating)			3	1.5%

Standard deviation (SD).

**Table 3 healthcare-11-01461-t003:** The dimensions’ items and total alpha.

No.	Items	The Alpha Coefficient
1.	I am concerned that my family and colleagues will avoid me due to my reinfection.	0.87
2.	Because I feared that someone would know that I had been reinfected with COVID-19, I avoided medical care.	0.86
3.	I am afraid that if I get COVID-19 again, I will pass away unexpectedly.	0.88
4.	Being re-infected with COVID-19 is what terrifies me the most.	0.82
5.	I believe it is vital to conceal from others the fact that I have been reinfected.	0.78
6.	I have recurrent nightmares in which I recall my previous COVID-19 infection.	0.93
7.	When I think of COVID-19 reinfection, I get a bad feeling.	0.80
8.	I worry that my COVID-19 reinfection may cause me to lose my job.	0.92
9.	When I worry about getting COVID-19 again, my heart beats fast.	0.83
10.	The reinfection with COVID-19 makes me feel inferior to others.	0.90
11.	Having COVID-19 again threatens my family’s reputation.	0.76
12.	It is uncomfortable for me to consider reinfection with COVID-19.	0.82
13.	If I am re-infected with COVID-19, I will feel embarrassed and ashamed.	0.86
14.	When I watch the news and read stories about the possibility of a COVID-19 reinfection, I become scared.	0.95
	Total	0.86

**Table 4 healthcare-11-01461-t004:** Pearson correlation coefficients of the first- and second-time participant answers with the overall score for each item.

No.	Items	R1	*p*	R2	*p*
1.	I am concerned that my family and colleagues will avoid me due to my reinfection.	0.46	<0.001 **	0.85	<0.001 **
2.	Because I feared that someone would know that I had been reinfected with COVID-19, I avoided medical care.	0.56	<0.001 **	0.88	<0.001 **
3.	I am afraid that if I get COVID-19 again, I will pass away unexpectedly.	0.42	0.003 **	0.80	<0.001 **
4.	Being re-infected with the COVID-19 is what terrifies me the most.	0.44	<0.001 *	0.84	<0.001 **
5.	I believe it is vital to conceal from others the fact that I have been reinfected.	0.37	0.004 *	0.76	<0.001 **
6.	I have recurrent nightmares in which I recall my previous COVID-19 infection.	0.50	<0.001 **	0.93	<0.001 **
7.	When I think of COVID-19, I get a bad feeling.	0.38	0.01 *	0.71	<0.001 **
8.	I worry that my COVID-19 reinfection may cause me to lose my job.	0.62	<0.001 **	0.87	<0.001 **
9.	When I worry about getting COVID-19 again, my heart beats fast.	0.54	<0.001 **	0.76	<0.001 **
10.	The reinfection with COVID-19 makes me feel inferior to others.	0.48	<0.001 **	0.78	<0.001 **
11.	Having COVID-19 again threatens my family’s reputation.	0.46	<0.001 **	0.74	<0.001 **
12.	It is uncomfortable for me to consider reinfection with COVID-19.	0.48	<0.001 **	0.82	<0.001 **
13.	If I am re-infected with COVID-19, I will feel embarrassed and ashamed.	0.51	<0.001 **	0.89	<0.001 **
14.	When I watch the news and read stories about the possibility of a COVID-19 reinfection, I become scared.	0.64	<0.001 **	0.90	<0.001 **
	Total			0.82	<0.001 **

R1: Pearson’s correlation of each item of the scale with total score. R2: Pearson’s correlation between the first- and second-time student answers. * Significant (*p* value < 0.05). ** Significant (*p* < 0.001).

**Table 5 healthcare-11-01461-t005:** Rotated factor analysis of FSoCOVID-19RS (*N* = 190).

No.	FSOCOVID-19RS Items	Factor 1	Factor 2
1.	I am concerned that my family and colleagues will avoid me due to my reinfection.	0.398	**0.786**
2.	Because I feared that someone would know that I had been reinfected with COVID-19, I avoided medical care.	0.536	**0.662**
3.	I am afraid that if I get COVID-19 again, I will pass away unexpectedly.	**0.644**	0.523
4.	Being re-infected with the COVID-19 is what terrifies me the most.	**0.795**	0.351
5.	I believe it is vital to conceal from others the fact that I have been reinfected.	0.446	**0.612**
6.	I have recurrent nightmares in which I recall my previous COVID-19 infection.	**0.591**	0.531
7.	When I think of COVID-19, I get a bad feeling.	**0.817**	0.362
8.	I worry that my COVID-19 reinfection may cause me to lose my job.	0.312	**0.792**
9.	When I worry about getting COVID-19 again, my heart beats fast.	**0.748**	0.267
10.	The reinfectionof COVID-19 makes me feel inferior to others.	0.240	**0.792**
11.	Having COVID-19 again threatens my family’s reputation.	0.453	**0.567**
12.	It is uncomfortable for me to consider reinfection with COVID-19.	**0.856**	0.318
13.	If I am re-infected with COVID-19, I will feel embarrassed and ashamed.	0.248	**0.825**
14.	When I watch the news and read stories about the possibility of a COVID-19 reinfection, I become scared.	**0.807**	0.312
	Total of eigenvalues	9.720	1.140
	Variance (%)	72.220%	9.045
	Cumulative (%)	72.220%	80.250%

Method of extraction: principal component analysis and method of rotation: Varimax with Kaiser normalization.

**Table 6 healthcare-11-01461-t006:** Convergent and discriminant validity of the FSoCOVID-19RS scale.

FSoCOVID-19RS Subscale	CR	AVE	MSV	ASV	1	2
Fear of reinfection	0.820	0.537	0.167	0.097	* 0.733	
Stigma of reinfection	0.835	0.549	0.207	0.106	** 0.226	* 0.741

* Square root of AVE values. ** *p* < 0.001 (two-tailed).

## Data Availability

The authors will provide this work’s datasets upon reasonable request.

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
