# Peer review of "Fear and Stigma of COVID-19 Reinfection Scale (FSoCOVID-19RS): New Scale Development and Validation"

_healthcare, 2023, doi:10.3390/healthcare11101461_

Round 1
Reviewer 1 Report
See attached

Requires a thorough proofread
Author Response
Thanks for your response and for giving us the chance to revise our manuscript.

Reviewer 2 Report
Thank you very much for inviting me to review this publication. The subject matter is very interesting. In the attached document are the considerations made after reading and reviewing the manuscript.
Best regards.

Author Response

(The authors gave the same response as above.)

Round 2
Reviewer 1 Report
The authors have addressed the minor concerns I had. However, I am still not convinced that this is a single scale but is of the view that these are two distinct and separate scales. However, the authors have provided reasons and responded to these issues and I must respect they have an alternative view on this.
Regards
Author Response
We would like to thank you for taking the necessary time and effort to review the manuscript. We sincerely appreciate all your valuable comments and suggestions, which helped us improve the quality of the manuscript. We would like to take this opportunity to thank you for the effort and expertise that you contributed towards reviewing our manuscript.

Reviewer 2 Report
The answers given are correct and well formed, however, there are a number of issues that remain unresolved.
I suggest you look for indexed terms, as it is very important that the keywords are correctly indexed so that the language is standardized to identify the topics correctly and universally.
Sorry for the insistence, you should indicate the total number of students in the nursing school and make a sample calculation.
Please make the changes I indicate.
Thank you very much,
Yours sincerely.
Author Response

(The authors gave the same response as above.)
